# Cadaver Dogs and the Deathly Hallows—A Survey and Literature Review on Selection and Training Procedure

**DOI:** 10.3390/ani10071219

**Published:** 2020-07-17

**Authors:** Clément Martin, Claire Diederich, François Verheggen

**Affiliations:** 1TERRA, Gembloux Agro-Bio Tech, University of Liège, Avenue de la Faculté 2B, 5030 Gembloux, Belgium; cmartin@uliege.be; 2Namur Research Institute for Life Sciences, University of Namur, Rue de Bruxelles 61, B-5000 Namur, Belgium; claire.diederich@unamur.be

**Keywords:** human remains detection dogs, dog behavior, dog selection, dog training, forensic science

## Abstract

**Simple Summary:**

Dogs are used in many contexts, including the detection of odor sources. A particular application of dogs during searching activities is the location of human remains. In this work, we aimed to gather the common practices of police dog handlers based on a survey made of nine questions, which was sent to police dog handlers around the world (N_Countries_ = 10; N_Brigades_ = 16; N_Handlers_ = 50) and then compared to the information available in the scientific literature. We highlighted a wide diversity of selection and training procedures used by handlers. Studies dedicated to human remains detection dogs are not abundant. However, we found key information that should be applied by handlers during the selection and the training of their human remains detection dogs. First of all, they should include the anatomical traits during the dog selection as behavioral traits alone are not representative of their potential. Finally, even if the training procedures are well performed by handlers, we highlighted a wide diversity of homemade training aids. However, no information was found in the literature regarding the effect of the training aids on the dog performances. For these reasons, handlers should create normalized selection and training procedures while scientists should investigate the behavior of these dogs to provide more information to handlers.

**Abstract:**

Human remains detection dogs (HRDDs) are powerful police assets to locate a corpse. However, the methods used to select and train them are as diverse as the number of countries with such a canine brigade. First, a survey sent to human remains searching brigades (N_countries_ = 10; N_Brigades_ = 16; N_Handlers_ = 50; N_questions_ = 9), to collect their working habits confirmed the lack of optimized selection and training procedures. Second, a literature review was performed in order to outline the strengths and shortcomings of HRDDs training. A comparison between the scientific knowledge and the common practices used by HRDDs brigade was then conducted focusing on HRDDs selection and training procedures. We highlighted that HRDD handlers select their dogs by focusing on behavioral traits while neglecting anatomical features, which have been shown to be important. Most HRDD handlers reported to use a reward-based training, which is in accordance with training literature for dogs. Training aids should be representative of the odor target to allow a dog to reach optimal performances. The survey highlighted the wide diversity of homemade training aids, and the need to optimize their composition. In the present document, key research topics to improve HRDD works are also provided.

## 1. Introduction

Human remains detection dogs (HRDDs) are “canines specially trained to find human decomposition scent and alert their handler to its location” [1]. In addition to be called cadaver dogs, HRDDs are most likely less known than other categories of working scent-detection dogs such as explosive or drug detection dogs. However, they are used by law enforcements in many contexts including homicides, as well as natural and manmade disasters to search for human cadavers, body parts, or fluids [1,2,3,4,5,6,7]. These activities are usually gathered under the term necrosearch [8].

During dog domestication, humans selected individuals upon their olfactory abilities, which were particularly important to the development of hunting dog breeds [9]. As a result, today’s breeds of working dogs show accurate, sensitive, and reliable olfactory abilities [6,10,11,12,13,14]. Dogs are able to recognize several hundred compounds due to a wide diversity of olfactory receptor cells present in their olfactory epithelium [9,15,16,17,18]. In addition, polymorphism in olfactory receptors genes increases the number of molecules that can be bound, making dogs powerful biological detectors. Not all dogs are suitable to perform detection work, as a result of e.g., their head conformation, their olfactory sensibility, or the polymorphism of their olfactory receptor genes [9,10,14,15,16,19,20].

The diversified environments in which HRDDs are deployed require dogs with specific skills. While some of them can be learned, several intrinsic traits would ease the training program. Handlers would benefit from identifying morphological, olfactory, and behavioral traits that indicate a dog’s suitability to become an HRDD. However, poor information is available regarding the selection of HRDDs, including traits and procedures. The scientific literature mainly focuses on the selection of service dogs [21,22]. As regards to the training procedures, dogs as pets are more studied compared to working dogs [23,24,25].

The conditioning of detection dogs relies on the quality of the available training aids [26]. HRDDs should be trained with aids that mimic the smell of any human cadaver. Some training aids are available on the market but were shown to be unrepresentative of the smell of a real decaying corpse [27]. The characterizations of the volatile organic compounds released by human corpses are scarce as they are complicated to carry out [28,29]. The cadaveric smell is composed of up to 800 molecules belonging to almost all chemical families including alkanes, ketones, aromatics, amines, and sulfur compounds [28,30,31,32]. Moreover, the decomposition of a corpse involves a wide range of interweave mechanisms making it highly variable and uneasy to understand [29,33,34,35,36]. Numerous biotic (e.g., tissues nature, animal species, necrophagous insects, microorganisms) and abiotic factors (e.g., humidity, temperature, death location) affect the volatilome of a cadaver [36,37,38,39,40]. In addition, a decaying corpse typically goes through five stages of decomposition respectively named fresh, bloated, active decay, advanced decay and dry remains, all characterized by different blends of volatile compounds [28,34,41]. As a result, the volatile chemical cues released during the decomposition process differ greatly from one case to another [4,28,34,36,38,42], making it difficult to provide HRDDs’ handlers with training aids mimicking human decomposition.

This review aimed to gather all available information on the selection and the training procedures of detection dogs. This research was carried out through two approaches: (1) We developed a survey that was sent to human remains searching brigades across the world, in order to collect their handlers’ working habits; (2) a literature review was carried out to outline strengths and shortcomings of the selection (anatomical and behavioral characteristics) and training of HRDDs. A comparison between the scientific knowledge and common practices used by law enforcement forces was then performed. Finally, based on this comparison, we have provided a guideline compiling the best scientific-based practices to improve HRDD works, as well as perspectives for further scientific researches.

## 2. Methods

Bibliographic research—The bibliographic research was performed using Google Scholar^®^, Science Direct^®^ and Scopus^®^ between June and December 2019. A first research was performed by focusing on the selection and training of cadaver dogs. The words’ combination used was the following: “(“cadaver dog” OR HRDD OR “human remains searching dog” OR “human remains detection dog”) AND (selection OR training)”. This research only highlighted two publications. We therefore decided to extend our research to other types of detection dogs and gathered information that could be applied to HRDDs. We ended up conducting different bibliographic researches according to the section of the present review (Table 1).

Regarding the ‘anatomical traits’ section of this work, we have extended the scope to wildlife detection dogs, because they have to face similar working environments as HRDDs: they mostly perform their searching activities outdoor, rather than indoor. Regarding the ‘methods of selection’, ‘behavioral traits’ and ‘olfactory traits’ section of this work, we have extended the scope to all detection dogs, because they are independent of the environment where the search is performed. Regarding the ‘training methods’ section of this work, all working dogs were taken into account as the methods can be considered equivalent. Finally, the ‘training aids’ section focused only on HRDDs because the olfactory cues used during the training have to be representative of the smell of a decaying cadaver.

Survey—A survey made of nine questions based on information from the literature and Belgian HRDD handlers (Table 2) was sent to the Kynopol Secretariat, who forwarded it to all affiliated human remains searching brigades around the world (LimeSurvey^®^). The survey was written in English. Sixteen brigades answered the survey between 1 August and 20 September 2019. Response to the questions was not mandatory. The handlers belonging to the brigades also had the possibility to add comments for each question listed in Table 2. Ten brigades out of 16 also described the HRDDs working in their brigades. Only the responses provided by HRDD’s handlers were taken into account. Fifty handlers from ten countries answered the survey, including Canada, United-Kingdom, Portugal, Belgium, Sweden, Finland, Austria, Slovenia, Romania, and Cyprus.

Through the present document, the word ‘brigade’ is applied to the entire department (including handlers and HRDDs). The results of the survey are presented at the beginning of each part of the review (selection and training).

Data analyses—A descriptive analysis of the survey responses has been performed. A subjective number between 0 and 100 was dedicated to each level of importance (not important = 0; important = 50 and very important = 100). We attributed a score to each selection parameters based on the answer of all handlers (Equation (1)). All the other answers were plotted by using Microsoft Excel (Version 16) or R Studio (RStudio, 2019). When possible, a chi-square test was performed to compare the proportions of response. However, due to a low sample size, statistical analyses were not often implemented.

Equation (1): Calculation of the score of importance of the different selection parameters evaluated in the survey.
(1)$$\mathrm{Importance}\;\mathrm{score}=\frac{\left(n_{\mathrm{not}\;\mathrm{important}}\times 0\right)+\left(n_{\mathrm{important}}\times 50\right)+\left(n_{\mathrm{very}\;\mathrm{important}}\times 100\right)}{N_{\mathrm{handlers}}},.$$
where n = number of handlers choosing a specific level of importance (not important, important, very important) and N = total number of handlers responding to the survey.

## 3. Selection of Human Remains Detection Dogs

Results of the survey—The survey recorded a number *of* 6.8 ± 2.0 HRDDs per country (data from seven countries). Moreover, based on our survey (N_Brigades_ = 16, N_Handlers_ = 50), we conclude that there is no standard procedure being applied for selecting dogs within a litter. All teams (except one) declared to implement homemade selection tests to assess the learning abilities of puppies during preliminary training sessions, taking into account some specific traits (e.g., drive, independence). In most cases, the selection is based on the handler’s own opinion during training or playing time only, without collecting any quantitative measures (“gut” feeling). Only two brigades (out of 16) perform normalized tests whose detailed description has not been provided. These two brigades assess the dog morphology as well as about 20 behavioral traits including what they call drives, courage, aggression, and nerve strength. Respondents mostly ignore morphological traits during the selection of HRDDs. They usually select puppies within the same breed (Figure 1). The four preferred breeds are: Malinois shepherd, German shepherd, English spaniel and Labrador retriever. Agility and stamina are the most important traits according to handlers’ responses, while dog size, fur length, and muscle mass are barely taken into account (Figure 2). Handlers from only two brigades pointed out that size and fur length are important traits that could influence dog performances. These two brigades mentioned that the perfect dog size is medium to tall while fur length has to be long. Among the brigades describing their HRDDs teams (10 brigades out of 16), males are more often used than females (χ ^2^ = 5.23; *p* = 0.02). In addition, they prefer to not neuter their dogs (χ ^2^ = 16.09; *p* < 0.001) with only 23% of HRDDs being neutered. All Canadian and English dogs are neutered. Neutered dogs are always males, no HRDD females are mentioned.

None of the 50 questioned handlers select their dogs based on olfactory traits. Most handlers agreed about the importance of taking behavioral traits into account during puppies’ selection. Handlers highlighted several important parameters: playfulness, curiosity, sociability, independence, and dynamism, with a score higher than 70. In addition, HRDDs should not lead the search and should follow the instruction of the handlers. Some handlers judged that a pronounced leadership is important for an HRDD. None of the surveyed handlers mentioned the importance to evaluate the ability of the dog to cope with stressful situations (Figure 2). Among the responding handlers, 36% assessed the drive during the selection.

Puppy selection remains crucial to obtain efficient adult HRDDs, as all dogs within the same breed or litter do not have the intrinsic qualities to perform such a complex work [9,14,16,43,44,45,46,47]. For this reason, a list of recruitment criteria must be defined. However, the survey reveals that no standard procedure is applied to choose dogs within a litter. Moreover, handlers mainly focus their selection on the behavioral traits while literature research suggests that physical characteristics are also important features to consider during the selection of detection dogs [20,22,45,48,49]. In the following paragraphs, we discuss the importance of four groups of features on the selection of a detection dog including (i) morphological traits, (ii) olfactory traits, (iii) behavioral traits, and (iv) the documented methods used to assess these three groups of traits.

Morphological traits—HRDDs perform search of decaying human remains in a large diversity of environments, including urban (e.g., collapsed building, garden) and wild environments (e.g., woodlands, grasslands, mountains, beaches) [2,5,33,50]. Just like wildlife detection dogs, HRDDs should be agile to move easily in rough fields and have steady stamina to stay performant during a long period of time [1,51]. Because the dog’s size (also called “dog body shape”) and the coat length are both correlated with dog’s mobility, they should both be considered during dog selection [45,52,53]. Three types of build (large, medium, and small) and two types of coat length (short and long) are typically highlighted in the literature [19]. Handlers should prefer a medium built dog (e.g., Springer spaniel, Labrador retriever, German shepherd) to ease their move in a rough environment. Another advantage of these breeds is that their thermoregulation is more efficient in various environments (e.g., warm/cold conditions, rough field) [47,53]. Among medium-sized dogs, breeds having proportionally longer legs show better agility [52,54]. Large dogs may experience trouble to cool down in hot or strenuous environments whereas small dogs are known to get tired faster under these environments [20,53,54]. Heat tolerance is not only regulated by the size of the dog but also by the length of the coat. A long coat will reduce the dog’s mobility while it will be an obstacle to the smooth running of his work under hot weather. On the other hand, a short coat decreases the performance of the dog in case of cold conditions. The main point to focus on is the ability of the dog to regulate its own temperature during the searching time as it will impact its stamina [20,52,53,55]. In their book ”Cadaver Dog Handbook”, Rebman et al. advise that the working environment should define dog size and coat length [1,19].

While agility was classified as the most important trait, size and coat length were not being considered by the international teams having answered our survey when selecting HRDDs. Sex has not been identified to impact HRDD’s performances. However, females seem to be less aggressive than males, suggesting that they would be proficient specialist searching dog [56]. However, our survey suggest that males are preferred over females. In addition, no differences were highlighted between neutered and unneutered dogs [56]. However, neutered dogs live longer than the non-neutered ones [57]. Finally, health and expected longevity of the dog have been identified as important factors, among others to not waste time with dogs that would not be able to work properly because of repeated sickness periods [43,45,58,59].

Olfactory traits—Some breeds have developed a more efficient olfactory apparatus than others, as a result of hundreds of years of selection [14,15,16,60]. The efficiency of the olfaction sense is linked to the structure of the dog’s skull. Three types of dog skull conformation can be described: brachycephalic, mesaticephalic, and dolichocephalic breeds (Figure 3). Although the exact description of these conformations is not clearly explained in the literature, researchers calculate the ratio between skull width and skull length, leading to average values of 0.81 for brachycephalic, 0.52 for mesaticephalic, and 0.39 for dolichocephalic [48]. The skull conformation will impact the orientation of the olfactory bulb and the volume of the nasal cavity. Brachycephalic breeds are characterized by a compressed nasal cavity, positioning their olfactory bulb in a ventral orientation and reducing the epithelial surface available for the capture of odors [48]. Brachycephalic breeds should, therefore, be avoided for olfaction-based work, and mesaticephalic and dolichocephalic breeds should be preferred [45,48]. The head conformation is a characteristic that should be taken into account to easily assess the olfactory capability of a dog: the larger the nasal cavity, the more olfactory receptors cells (ORCs) are located in the epithelium [61,62]. During our survey, questioned handlers did not mention the structure of the head as a selection factor. However, no brachycephalic breed is used by the questioned teams (Figure 3). To be perceived by the dog’s brain, odorant molecules must enter the nasal cavity and reach the olfactory epithelium where olfactory receptor cells (ORCs) bind them. The number of ORCs, which varies among dog breeds, is therefore another important parameter to consider. For example, bloodhounds have 300 million ORCs (the higher number among dog breeds), and German shepherds have 225 million ORCs [61,62]. Variations are observed among dog breeds but also among individuals within the same breed [14,46,56]. Almost 1100 genes are responsible for ORCs synthesis [9,14], which makes any selection effort based on genetic characteristics a difficult task [14,15,16,60]. No genetic selection was mentioned in additional comments, suggesting that they base their selection on interesting genes indicating a potential future performant HRDD. A third olfaction-associated parameter that should be considered when selecting a potential HRDD is the size of the ear and dewlaps. Large ears and dewlaps allow a dog to catch more volatile molecules focusing the scent around the nose [19]. The size of the ear and dewlaps were not mentioned in the answers of the survey. Yet, German shepherd dogs—which have a dewlap—and English spaniel dogs—which have large ears—are among the most common HRDDs (Figure 1).

Behavioral traits—Behavioral traits also play critical roles in the determination of the success or failure of future HRDDs’ training [19,61,63]. The behavioral response of a dog facing a particular stimulus is driven by two main components: its temperament and its personality [21,64,65,66]. Temperament can be defined as the individual difference in behavioral responses, which is steady among time and context, that are grounded in affective state and its regulatory processes and which are evident from an early age [67]. It usually has genetic predisposition [58,63,67,68]. Personality is the result of both a dog’s genetics and experiences over time, which shapes the dog’s behavioral response to situational events [19]. Two main behavioral traits should be considered in HRDDs: the drives and the nerve strength [22,45,69,70].

Some authors consider the term “drive” as synonymous to motivation [43,58,68,70] which is defined as an innate impulse that prompts a canine action while others make the distinction between drive and motivation but without explaining the differences [19,45,47,69,70]. However, drive is a generic term that includes at least six sub-categories, according to the available literature: (i) pack or social drive, (ii) play drive, (iii) food drive, (iv) prey drive, (v) hunt drive, and (vi) defense drive [63,69]. Pack drive is the ability of the dog to work in cooperation with humans [45]. Play drive is the desire to be entertained [45]. Food drive is the desire to obtain food [69]. Prey drive is defined as the willingness to engage in a competition game [22]. Hunt drive is the ability to hunt and locate a prey, and also comes with a desire to be rewarded [71,72]. Defense drive is the tendency to defend itself or its handler [22]. Most of the many behavioral traits listed in the literature (such as trainability, personality, stress, fearfulness, and courage) could be related to one of the drives previously described [73]. In addition, drive is linked to the concentration of the dogs, i.e., its ability to remain focused on its searching work despite the presence of distracting scents or events [52,63]. Drive is among the most important behavioral traits to consider during the selection of a scent detection dog according to the scientific literature [69,74,75,76]. Indeed, the stronger drive is, the faster the training is and the better the final performances of HRDDs are [19,52,63]. Scientists however consider that some drive types are more important than others [63]: Play drive is one of the most important for any wildlife detection dog, including HRDDs. A high play drive dog will have a strong desire to train and receive its reward, which is essential for training and work [1,45,52]. Later, the dog will engage more easily with the searching work. A high play drive dog will be less distracted during the training and the work. Indeed, the dog will concentrate on getting the reward (i.e., toy) [19]. Food drive is most likely as important as play drive when it comes to HRDD’s selection. Indeed, using a food reward is supposed to bring similar result than a toy [63,69]. We suggest to gather play and food drives under a single drive category: reward drive. The reward drive would be defined as the desire of the dog to get the reward. The more the dog wants its reward, the more it will be focused on its work and the less it will be distracted by other stimuli. Two other drives should be considered by handlers: hunt and prey drives [46,69]. Hunt drive will inform the handler on the motivation of the dog to locate a target without perceiving it by using its nose. It is strongly linked to the prey drive. However, some scientists highlighted that a too high prey drive can be catastrophic during the location of a victim: these dogs could attack the target when they find it (which is not desired during the search of decaying human bodies) [74]. One last important drive is the pack drive or social drive: detection dogs have to be able to work with unfamiliar dogs and humans without expressing any anxiety or fear. Pack drive also gives an idea of the level of cooperation that can be expected between a dog and its handler. No study evaluated the impact of high pack drive level on HRDD performances. However, we can hypothesize that the better the ability of a dog to understand the gestures and orders of its handler (which is linked to the cooperation with the handler), the easier will be the training [52,77]. While a highly developed defense drive is a prerequisite for most police working dogs, it is not a required trait for detection dogs [69].

The nerve strength is another important behavioral trait to consider. The nerve strength is the ability of the dog to deal with a stressful situation, such as a noisy sound [19,69]. This parameter has to be specially considered for HRDDs as they are likely to work under highly stressful environment [1]. Indeed, dogs exposed to stressful environment will reduce their activity level and are less efficient during learning session [78]. HRDDs are trained to not react to auditory, tactile and visual stimuli, as they are likely to work in disaster environments. Indeed, they must be able to work on unstable ground surfaces, tunnel, or crawl spaces. They are likely to be exposed to gunshots, rock collapse, and fire smoke [52,69]. However, it was not confirmed by scientific research that the presence of such stimuli impact the dog’s performances [63,69].

Handlers mentioned the drive in the additional comments. Questioned handlers never mentioned the different types of drive in the survey, even if based on the literature review, the drive must be evaluated. Yet, as previously mentioned, none of the surveyed handlers detailed the drives they evaluated. However, they did evaluate some behavioral characteristics that may be related to the drive, such as playfulness, sociability, and curiosity. These traits can be easily associated with play drive, pack drive, and prey or hunt drives. During our review of the literature, we had to face the lack of homogeneity within the vocabulary used to describe behavioral traits which explain the wide diversity of terms used to describe a unique behavior. On the other hand, some handlers judged that a pronounced dominant trait is important for an HRDD, which is not in accordance with the literature that states that a detection dog should always be led by its handler for more efficient results [45,52,77]. Finally, none of the surveyed handlers mentioned the importance to evaluate the ability of a dog to cope with stressful situations, suggesting that they most likely do not evaluate this ability. They should however train their dog to better resist stressful situations. Indeed, repeated exposure to stressful stimuli increase nerve strength [63,69,79]. However, dogs with a high food or play drive may have a higher nerve strength, since they are likely to keep focused on their target to receive their reward as soon as possible. As food and play drive are taken into account during the selection of HRDDs, they might select, simultaneously, higher nerve strength dogs [1]. Standardized tests would be helpful for a more accurate evaluation of behavioral scent detection dog traits, including HRDDs, during their selection.

Selection procedures—Several methods of working dog selection are described in the literature and are summarized in Table 3. Most of these studies deal with the selection of service dogs [21,22,80]. Even if several traits are important for both service and detection dogs, different works require different skills. For instance, while a guide dog should constantly stay close to its handler and keep looking at him to receive its instructions, a detection dog works at a distance and pays attention to its handler’s verbal communication [80,81]. Most of these methods assess nerve strength and the above described drive types [19,45].

Two of these selection procedures were scientifically assessed: based on C-Barq assessment (survey on the personality of the dog), Reference [82] and on IFT (In-For-Training, exercises or tests, [80]). Both procedures were shown to be accurate and improved the selection of service dogs [65]. However, the evaluation of these selection procedures did not document scent detection dogs. They were based on the observation of an evaluator, breeder or handler, making the tests subjective. In addition, no metrics were provided to decide after the selection procedure which dogs should be selected. Finally, these documented procedures mostly focus on behavioral traits without taking anatomical and olfactory traits into account. To validate a selection method, it is required to demonstrate its predictive value (i.e., performance of the detection dog). However, no performance assessment tests and skill measurements were found in the literature for HRDDs, most likely explaining why handlers continue to perform homemade selection procedures. We strongly recommend their developments. This lack of interest in the scientific community could be explained by the small number of HRDDs in each country, as revealed by our survey which shows that 6.8 ± 2.0 HRDDs are operational per country (data from seven countries). A common agreement on the terms used to describe the behavioral traits has to be adopted as well. These recommendations should help to standardize the selection of HRDDs and improve their final performances. Finally, the dog genetic (through genetic markers) could also be considered during the selection, because morphological and behavioral traits are inherited [16,60]. The identification of specific markers would ease the selection and could help avoiding subjectivity during the selection.

## 4. Training of Human Remains Detection Dogs

Results of the survey—During their training sessions, most surveyed handlers declared using reinforcement rather than punishment (Figure 4). A minority of them, however, still mentioned the practice of punishments. When they do so, they do not give the reward to their dog (negative punishment). They are a minority to the handlers who use positive punishment (Figure 4). Among handlers practicing positive reinforcement, no generalization can be made. They reported to use either toys, clicker, food, or encouragement as positive stimuli. Most of them use a combination of the different rewards. Handlers train their dogs on average for 33.3 ± 3.2 min, 3.2 ± 0.4 times per week. Handlers did not explain the way their training sessions are organized. No information was provided about how the training session ends.

Stimuli used during the specialized olfactory training were very different among teams (Figure 5). They reported to only use homemade biological aids (human or swine origin). Most of the handlers used aids of human origin. Commercial aids available on the market were not used.

Training methods—Most of the research performed on dog training procedures is dedicated to pets [24,25]. Documented HRDDs’ training methods are scarce and when available, they focus on case studies [13]. Moreover, none of them were associated with an evaluation of the performances and behavioral outcomes of the dogs. Training methods were based on operant conditioning during which a dog learns that its response to a command has consequences. The dog will repeat the behavioral response to the specific stimulus that leads to the higher benefit [83,84]. The latter may increase aggressiveness and fear and should therefore be avoided [25,83,85]. Reinforcements are associated with improved abilities to learn [25,86]. Moreover, a positive reinforcement leads to the creation of a strong relationship between a dog and its handler. While negative reinforcement is associated with distraction and lower obedience [83]. Despite these observations, the type of reinforcement remains a matter of debate among handlers: it is still unclear whether aversive and reward-based methods lead to different levels of performance in working dogs [25].

End of session cues—in a recent publication [87], the Federal Bureau of Investigation (FBI) raised the importance of End of Session Cues (EoSC). EoSC is a (series of) stimulus that informs the dog about the coming end of the training session. EoSC must be clearly and repeatedly introduced at the end of each training session to avoid the dog to associate events that follow the training session (e.g., return in the cage) as part of the training. If EoSC are not applied, the dog could reinforce unwanted behavioral responses. For instance, going back to the cage immediately after a training session could be considered by the dog as a punishment, resulting in a decrease in its performances. Instead a training session should end in a positive way, for example a playing period, which will likely increase dogs’ performances [87,88].

Aids—There is a wide diversity of materials that can be used to train HRDDs, including biological aids and synthetic ones [27]. As far as the authors know, no publication comparing the efficiency of various training aids on dogs’ performances is available. So far, scientists seem to agree that the biological aids are the most efficient and reliable ones that can be used to train HRDDs [4,89]. However, biological materials are difficult to obtain due to ethics, legislation, and biohazard risk for both human and dogs [5,27,71]. As already mentioned, the chemical profile released by a cadaver was deeply investigated by research who highlighted their highly variability [28,33,34,38,40,42], making HRDD work hard to perform, and the choice of the training aid difficult to make. Dogs must be able to recognize a wide range of chemical compounds associated to cadavers that may be fresh or putrefied, entire or torn into pieces, buried or placed under open sky [71]. Gravesoil and cadavers’ clothes seem to be among the most representative sources of the smell of a decaying corpse, because they accumulate fluids and are imbued by cadaveric odors [4,90]. When they use cadaver clothes, handlers must pay attention to choose cotton clothes, because cotton better adsorbs cadaveric VOCs than composite tissues. Moreover, even if the clothes accumulate the odor released during the decomposition process, it is hardly recommended that the handler uses textiles associated with a different post-mortem interval to present to the dogs an overview of the different situation that can be met [4]. Regarding gravesoil as training aids, one should pay attention to use a control soil to avoid the dog to be trained on non-targeted chemical substances (related to soil, and not to a cadaver) [90]. The use of cadaver pieces as training aids (such as bones or flesh) should be avoided. They are not just difficult to obtain; they also do not cover the entire diversity of volatile chemicals released by a cadaver [28]. In addition, the use of cadaver pieces as aids make them unavailable for the court to perform additional analyses, and the preservation of forensic evidences can be compromised [91]. A solution could be the use of human surrogate model. However, animal decomposition leads to different volatile signatures among species. The cadaveric volatile compounds released by decaying pigs are, for instance, different from those released by decaying human bodies [36,92]. Among the most reliable solution of training aids lies the Scent Transfer Unit 100TM (STU-100), which allows to trap the scent of a decaying corpse by pulling the surrounding headspace air. The air passes through a gauze which traps the volatile molecules and can then be used as training aids. This technique does not compromise the integrity of the material (e.g., corpses, clothes, soil) and meets ethical and biohazard recommendations. It provides a reliable training aid because it is representative of a dead body, adsorbing 60 to 85% of the total post-mortem VOC profile [93]. Commercial aids, also called synthetics aids, are not representative and reliable, because they do not contain compounds that have been previously reported within the headspace of human decaying bodies [27].

Our survey revealed that handlers mostly use human origin aids which is in accordance with the scientific recommendations. However, they also meet biohazard and ethics problems. Those who use a human surrogate model to train their dogs have to face to the potential differences existing between these model and human corpses. None of the handlers mentioned the use of commercial aids. In our opinion, developing a reliable synthetic aid is a promising perspective that would ease the training efforts of HRDD brigades. It requires a fine characterization of human post-mortem volatilome, and the identification of its key components, common to any decomposing corpse [29]. HRDD olfaction should also be better understood to avoid including compounds that cannot be perceived by the dog olfactory system. We recommend further researches to identify the chemical compounds that can be perceived by HRDDs, and to provide the composition of a synthetic training aid that would be cheap, easy to make, and lead to high levels of performance [27,89,90,94].

## 5. Conclusions

In addition to the recommendations made above, we suggest some perspective of research and suggestions to the HRDD brigades.

First, there is a lack of validated methods to be applied regarding the selection of puppies. We suggest the existing methods of selection to be validated. A selection method based on anatomical, olfactory, and behavioral traits should be developed. As no quantifiable data were recorded during the selection performed by handlers, this method should include measurable data to avoid subjectivity and to allow dog handlers to select the most promising individuals. So far, no bio markers indicating a potential good detection dog are available. One should therefore investigate the possibility to use, during the dog selection, biomarkers associated with high dog’s performances. To validate methods or potential biomarkers, it is important to develop a protocol for measuring dog performances. As no consortium is dealing with the vocabulary used to describe behavioral traits of detection dogs, we also recommend a common agreement on the vocabulary to be used.

Regarding training procedures, one should compare the impact of conditioning methods (including the use of clickers) on HRDDs’ performances. The olfactory aids used during the training of HRDD are of prime importance. The impact of the composition of a HRDD training aid (biological or synthetic) on the dog performance has never been evaluated. In our opinion, the development of a synthetic aid mimicking the human cadaver volatilome is most likely one of the most promising perspective, as this would provide handlers with a training tool that is reliable, ethical, easy to obtain and use.

## Figures and Tables

**Figure 1 animals-10-01219-f001:**
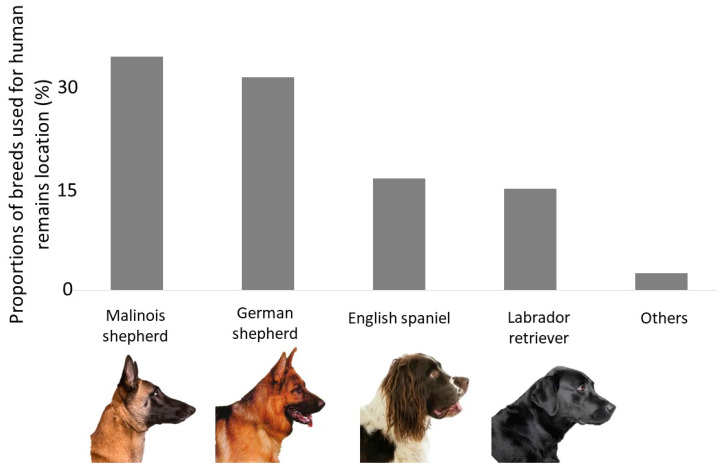
Breeds currently used as human remains detection dogs based on a survey completed by ten brigades from Europe and Canada.

**Figure 2 animals-10-01219-f002:**
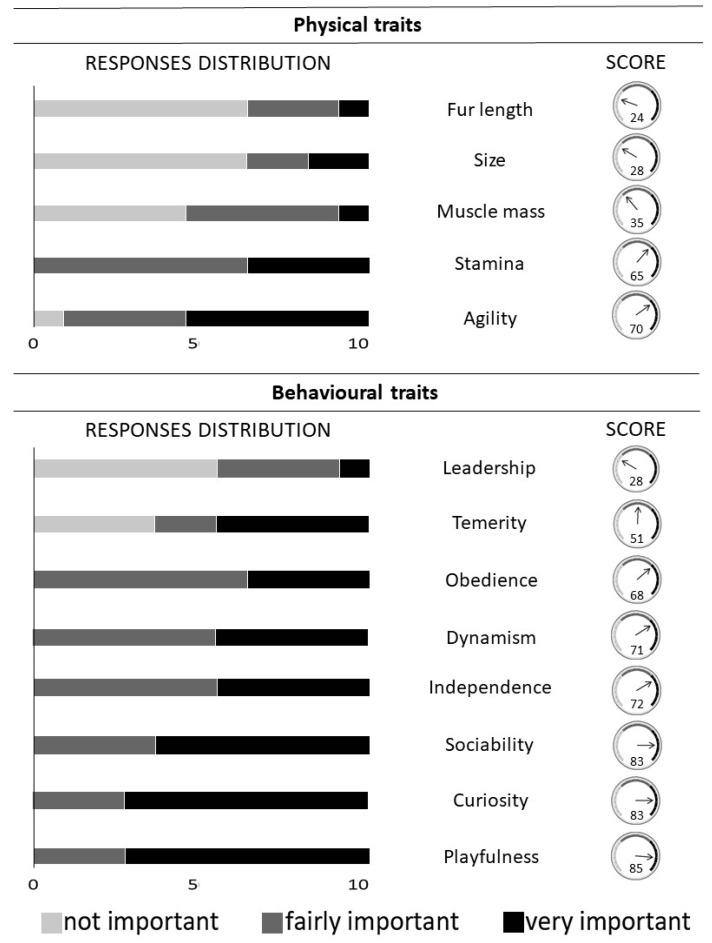
Handlers’ estimation of the importance of the physical and behavioral traits that must be considered during the selection of human remains detection dogs. The response distribution describes the percentage of handlers (out of 50) that estimate the importance of each trait (not important, fairly important, very important). The score estimates the importance of the traits during the selection according to Equation (1).

**Figure 3 animals-10-01219-f003:**
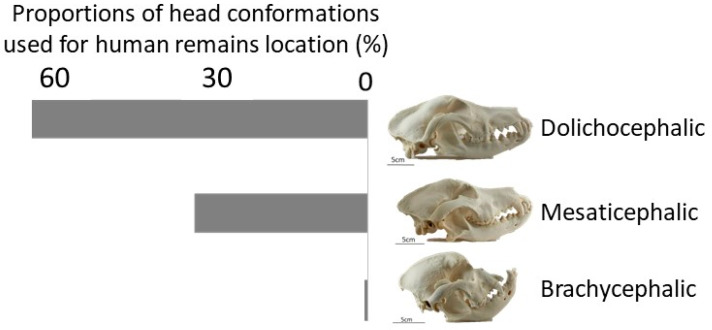
Proportions of dogs declared to be used as human remains detection dogs in our survey (N = 10 out of 16 brigades) belonging to the three common types of dog’s skull conformation (skull pictures are reproduced with the kind permission of Tibor Csörg).

**Figure 4 animals-10-01219-f004:**
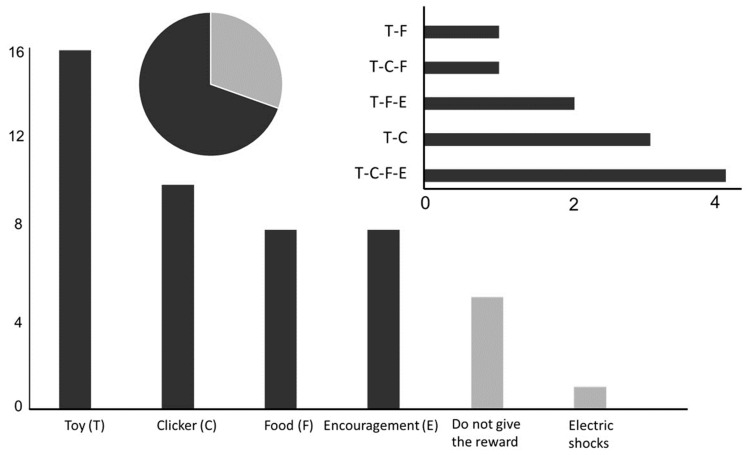
Types of operant conditioning used by the surveyed handlers (y axes, N = 50) to train their human remains detection dogs with the pie chart representing the proportion of reinforcement and punishment technics used (light grey: punishment, dark grey: reinforcement).

**Figure 5 animals-10-01219-f005:**
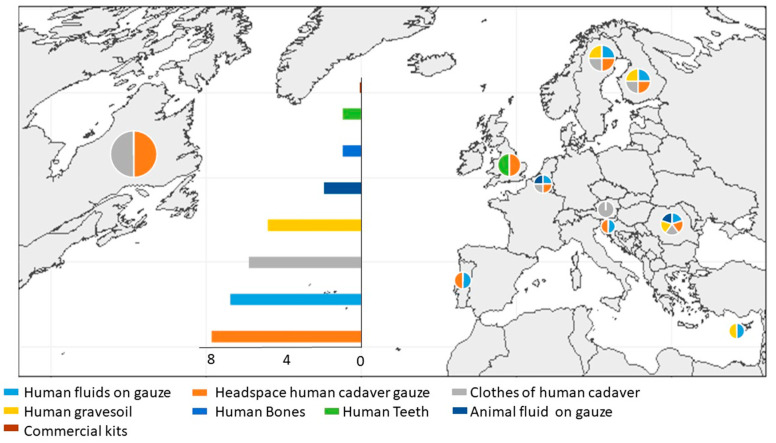
Types of training aids used by the surveyed human remains detection dogs’ brigades to train their dogs. Colors in the pie charts represent the diversity of training aids used by the teams in the country where the chart is located. The bar chart represents the number of human remains detection dogs’ teams (y axe) which use the different training aids.

**Table 1 animals-10-01219-t001:** Bibliographic research performed for each section of the present review using Scopus^®^, Science Direct^®^ and Google Scholar^®^ between June and December 2019.

Review Outline	Combinations of Keywords
**Dog Selection**	
Anatomical traits	Morphological: (selection AND suitable AND wildlife AND “detection dogs”)
	Olfactory: (selection AND “detection dog” AND ((olfaction OR “Olfactory system”) OR (“genetic marker”))
Behavioral traits	(selection AND (“detection dog” OR “scent detection dog”) AND “behavio(u)ral trait”)
Methods	(selection AND (method OR methodology) AND police AND forensic AND “detection dog” AND (method OR procedure OR methodology))
**Dog Training**	
Methods	(“human remains searching dog” OR “cadaver dog”) AND (“training procedure” OR “training method” OR “training methodology”)
Training aids	((“human remains searching dog” OR “cadaver dog”) AND “training aid”)

**Table 2 animals-10-01219-t002:** Survey sent to handlers (N = 50) of human remains detection dog (HRDDs).

**Survey Outline**	**Questions**
**Selection**	
Anatomical traits considered during the selection of HRDDs	Q1: What are the physical characteristics you are paying attention to when you select a dog to train? (for each trait, select among: very important/important/not important) Size Length and thickness of the fur Ability to run Muscle mass Q2: How do you assess the physical characteristics?
Behavioral traits considered during the selection of HRDDs	Q3: What are the behavioral characteristics you are paying attention to when you select your searching dogs? (for each trait, select among: very important/important/not important) Dominant Social Curious Dynamic Player Independent Obedient Adventurous Q4: How do you assess these behavioral characteristics?
**Training**	
	Q5: What kind of training do you perform? Punishment (including take off the toy). If yes: What kind of punishment? (electric shock/strike/other) Reward. If yes: Which kind of reward? (Toy/vocal encouragement/food/clicker/other) Q6: What is the duration of a training session (in minutes)? Q7: What kind of training aids do you use? Compress with human cadaveric fluids Compress introduced in a container with human organs and/or blood Compress with animal cadaveric fluids Clothes wear by human cadaver Gravesoil Commercial kit Q8: Could you describe your trainings? Q9: If you use biological training aids (compress with cadaveric fluid, gravesoil or clothes…), can you describe how you obtain the final trainings aids?

**Table 3 animals-10-01219-t003:** Selection methods of detection and service dogs.

Characteristics	Description (According to Authors)	Method				
		C-Barq	SDTC	IFT	M-B Scale	GDTP
**Drive**						
Causal reasoning	Use of visual and auditory cues to infer the location of hidden reward				∆	∆
Commitment to toy	Tendency to engage game with its handler (new toys or familiar toys)	∆			∆	∆
Retrieval inhibition	Ability to inhibit prepotent motor response in object retrieval task					∆
Energy level	Show enthusiasm and be always ready to play	∆				
Excitation	Before a walk or when owner/visitor is coming home	∆				
Gaze direction	Ability to use human gaze direction to locate hidden reward				∆	∆
Hiding-finding	Object permanence				∆	∆
Hunting behaviour	Tendency to track its prey directly or after a turnaround of 360°	∆		∆	∆	
Odour control trials	Control trials ruling out ability to locate hidden food using olfaction				∆	∆
Perspective-taking	Tendency to obey/disobey a command depending on whether a human is watching				∆	∆
Play with stranger	Tendency to play with familiar toy and stranger			∆	∆	
Reaching	Ability to infer reward location based on experimenter’s reaching towards baited location					∆
Retrieval	Tendency to retrieve object and return it to in front of experimenter	∆			∆	∆
Reward preference	Preference for food or toy reward					∆
Rotation	Egocentric vs. allocentric use of spatial cues					∆
Social referencing	Tendency to look at human face when joint social activity is interrupted				∆	∆
Spatial transpositions	Ability to track location of hidden reward across spatial transformations					∆
Visual discrimination	Ability to learn arbitrary visual discrimination prediction reward location					∆
**Nerve strength**						
Affect discrimination	Preference to approach unfamiliar human	∆	∆	∆	∆	∆
Confidence on rough surface	Ability to stay confident on rough surface				∆	
Confined space	Ability to come from a confined space and enter a lighted area and dark area				∆	
Fearfulness	Shaking/salivating/agitation/loss of appetite when they are left on its own	∆				
Laterality: first step	Forelimb preference when initiating a step off a platform	∆			∆	∆
Sociability towards other canines	Preference to approach unfamiliar canine	∆		∆	∆	
Sound sensitivity	Ability to stay confident when confronted to several sounds	∆	∆	∆	∆	
Stability	Ability to stay confident and stable on unstable surface				∆	
Surface sensitivity	Ability to travel across a slick surface				∆	
Threating situation	With stranger, unfamiliar dog or object		∆	∆		
Touch anxious	Tendency to stay calm when manipulated by human	∆		∆		
Visual sensitivity	Ability to stay relaxed and confident in an area full of smoke	∆			∆	
**Trainability**						
Arm pointing	Ability to use human arm pointing to find a hidden reward					∆
Detour navigation	Navigation of shortest route around an obstacle					∆
Inferential reasoning	Ability to infer the location of hidden reward through the principle of exclusion					∆
Marker cue	Ability to infer location of hidden reward when human uses a novel communicative marker					∆
Memory-distraction	Memory for location of reward across delays while dog’s attention is discarded					∆
Odour discrimination	Discrimination and memory for which of two locations is baited using olfaction					∆
Response to command	Ability to sit/stay	∆				
Spatial perseverations	Ability to inhibit previously established motor pattern when environment changes					∆
Working memory	Memory for location of reward across temporal delays					∆
**Personality**						
Attachment	Sign of attention to the owner	∆				
Begging for food	When people are eating	∆				
Contagious yawing	Tendency to yawn during auditory exposure to human yawning vs. control sounds					∆
Dynamism	Hyperactive or restless	∆				
Escapes	Take each opportunity to escape	∆				
Olfactory interest	Rolls when facing smelly substance	∆				
Owner direct aggression	Tendency to be aggressive with owners during daily tasks (batch, eating, game…)	∆				
Steal behaviour	Steal food	∆				
Unsolvable task	Help seeking from human vs. independent behaviour when facing unsolvable task					∆
**Morphological**						
Physical exam	Examination of the body tension			∆		
**Others**						
Barks	To alarm or when excited	∆				
Coprophagy	Eat its own faeces or of another animal	∆				
Licking	Itself or people or object	∆				
Pull on leash	During walk	∆				
Sensory bias	Prioritization of visual vs. olfactory information when senses pitted against one another					∆
Transparent obstacle	Ability to inhibit direct approach to experimenter when a detour is required					∆
Urinates	During night, when owner approaches, on object	∆				

∆ indicates that the method takes the characteristic into account; C-Barq (Canine Behavioral Assessment and Research Questionnaire; survey), IFT (In For Training; tests), SDTC (tests developed by the Swedish Dog Training Center), M-B scale (tests developed by Brownell and Marsolais 2000) and GDTP (tests developed by the Guide Dog Training Program) M-B scale is the only test having been performed on detection dogs [21,65,69,80,81].

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
