# Peer review of "Cadaver Dogs and the Deathly Hallows—A Survey and Literature Review on Selection and Training Procedure"

_animals, 2020, doi:10.3390/ani10071219_

Round 1
Reviewer 1 Report
Contribution to the Scientific Literature: This is an interesting topic in the field of working dogs, in an area where there is currently little scientific evidence. Selection of working dogs, including standardization of selection methods and criteria has a lot of interest from the working dog breeding and training industry, as well as from agencies that employ working dogs and readers from those disciplines may find this study interesting.
Format: In its current version, the manuscript tends to slip between the two formats that the authors have used: original research (survey study) and review of the literature and it becomes confusing at times to follow because of this. I don't feel that the results of their survey are clearly described in the abstract or the body of the manuscript, although you provide charts that help clarify your results. You have acquired quantifiable data with your survey, yet this is not provided in the abstract, and are difficult to find within the manuscript body. This could be improved by separating the results and discussion sections and first describing the results without commentary, then following with the discussion.
The number of respondents, teams, and other quantification of the study population for the survey was confusion and should be included in the abstract and clearly described in the Materials and Methods and Results Section, near the beginning. It isn't until line 237 that you state that the number of handlers was 50 from 9 different teams, and not clear if this was in response to a specific question or the total number of respondents.
Can you explain or describe how you know that your survey was sent to "all affiliated human remains searching teams around the world?" What criteria are you using to define "affiliated human remains searching teams?" What would exclusion criteria be in this case?
It would be helpful to define the word "team" as "canine team" can mean a paired handler and dog, or could also mean the organization which might have multiple paired handlers and dogs.
On line 124 you say:
"Based on our survey (n=16), we conclude that there is no standard procedure..." but then on line 237 you state " among the handlers who responded to our survey (n=50)..." I realize that these numbers may represent the total number of responses you received for that particular survey question, however it is not clear if this larger number (50) refers to the total number of respondents.
On lines 117-118, the authors suggest that there are only about 7 HRDDs in each country. In the united States, there are hundreds of HRDDs, and it is not clear how you came to this number, unless that is based on how many responses they received in your survey.
Type of Manuscript: Since a discussion of the available scientific literature is part of any manuscript of original research, I think this manuscript could focus on the survey study and not attempt to describe it as a "review of the literature," since the editorial and peer review criteria for a scientific review and original research are somewhat different. The authors could still provide ample discussion of the topic in their introduction and discussion sections without classifying the manuscript as a "review."
English Language: There are multiple minor errors that are likely related to translation:
Lines 150, 245, Figure 1: The authors use the word "research" in two areas where based on context it appears they are using this to mean "search" (the dog searching for target odor).
Line 26: "...police associates..." While technically this makes sense, "assets' may be a more appropriate term than "associates."
Lines 29, 54: The term "thank a..." is used where "due to" may be more understandable.
Citations: The authors tend to excessively cite review articles books, or other sources other than original research as scientific evidence to support their statements. Review articles and books should not be used in this manner, unless the authors are specifically providing commentary on trends within the topic, or commenting on the review authors' conclusions. In addition, you appear to have cited information provided as citations in the introduction or discussion section of original research papers (vs. citing the results or specific conclusions). I recommend that if you have supporting evidence that came from a previous review or introduction/discussion section citation, that you find the source of original research and cite that instead. This occurs in these locations within the manuscript: (Not completely inclusive)
Line 155 (19, 45)
Line 160 (19, 43)
Line 161 (19, 45)
Line 167 (19, 45, 53)
Line 168: (1, 19, 45)
Line 193: (45)
Line 199 (19)
Line 207 (19, 45)
Line 216 (19)
Line 219 (19, 22, 45)
Line 223 (43, 63)
Line 225 (19, 45, 61)
Line 284 (167)
333 (25)
Granted, there is little original research currently published on the topic, however as mentioned above, a true review of the scientific literature would not include review articles or books as sources of scientific evidence, although you night provide commentary on their quality or use them to direct readers to a source of compiled information on the topic. Thus, a true review of the scientific literature would highlight how much or little original research evidence is available. In this version of their manuscript, the authors are largely compiling non peer-reviewed informational publications and providing them as scientific evidence.
One more appropriate way of discussing these informational publications is to caveat statements and attribute the statement as an opinion of the original author. For example (Line 168): "The working environment should define the preferred dog size and coat length. [1,19,45]" could be written as "In their book Cadaver Dog Handbook, Rebman et al advise that the working environment should define dog size and coat length because... although we found no studies on this in the available scientific literature." Then provide commentary on why they agree or disagree with the opinion of the authors of the book.
Study Design: The survey seems to inhibit possible responses by limiting both physical and behavioral traits as well as training methods to a short list. I did not see an option for "other" as a potential response with ability to write-in responses other than "Can you describe your trainings?" which is a very broad and open question. I see this as a potential flaw in the study design. Can the authors elaborate on if the opportunity to respond outside of the provided options was available?
You describe expanding your search criteria to include wildlife dogs starting in line 142, when a decision on study design such as this should have been described in methodology vs. Results or Discussion. You could simply state in Methods that because there would likely be more literature on these other types of dogs, and since the characteristics would be similar, you included those types of dogs in your search.
Analysis of results and conclusions: I feel that the small sample size makes accurate and representative analysis difficult. Because it is unclear how many respondents you had and number of potential respondents in the target population, it is difficult to assess the overall quality of your results. Because you did get quantifiable results (number of respondents, importance score, etc.) I think should attempt a more robust statistical analysis than simply descriptive statistics.
Author Response
Referees comments in normal type – Our comments in bold
We are grateful for the careful reading and the many comments.
Reviewer #1
Format: In its current version, the manuscript tends to slip between the two formats that the authors have used: original research (survey study) and review of the literature and it becomes confusing at times to follow because of this. I don't feel that the results of their survey are clearly described in the abstract or the body of the manuscript, although you provide charts that help clarify your results. You have acquired quantifiable data with your survey, yet this is not provided in the abstract, and are difficult to find within the manuscript body. This could be improved by separating the results and discussion sections and first describing the results without commentary, then following with the discussion.
Answer: To deal with the comments of all three referees, we have decided to keep the original structure of the manuscript, and still consider this work as a review but to highlight the results of the survey at the beginning of each section. The survey allows the reader to understand the lack of communication between scientific research and working brigades.
The number of respondents, teams, and other quantification of the study population for the survey was confusion and should be included in the abstract and clearly described in the Materials and Methods and Results Section, near the beginning. It isn't until line 237 that you state that the number of handlers was 50 from 9 different teams, and not clear if this was in response to a specific question or the total number of respondents.
Answer: This information is now provided in the abstract (line 29-30) and in the material and method section (line 107-109).
Can you explain or describe how you know that your survey was sent to "all affiliated human remains searching teams around the world?" What criteria are you using to define "affiliated human remains searching teams?" What would exclusion criteria be in this case?
Answer: Kynopol is the European Police network for law enforcement dog professionals which is part of CEPOL (the European Union agency for law enforcement training). It gathers all dog handlers, including detection dog handlers, in Europe and they also have contact with other countries around the world. As all countries in Europe do not have human remains detection dogs, all countries did not respond to our survey.
The only exclusion criterium is that the handler answering the survey has to be a HRDD’s handler. All other detection dog’s handlers were not taken into account.
For more clarity, we now mention the name of all countries having answered to the survey, even if more than one brigade (made of several dog-handler teams) is present in the country.
It would be helpful to define the word "team" as "canine team" can mean a paired handler and dog, or could also mean the organization which might have multiple paired handlers and dogs.
Answer: We now define the words “brigades” and “canine Team” in the material and method (line 109-111)
On line 124 you say: "Based on our survey (n=16), we conclude that there is no standard procedure..." but then on line 237 you state " among the handlers who responded to our survey (n=50)..." I realize that these numbers may represent the total number of responses you received for that particular survey question, however it is not clear if this larger number (50) refers to the total number of respondents.
Answer: We make it clearer in the material and method section (line 107-111). The survey was answered by 50 handlers from 10 countries and 16 brigades
On lines 117-118, the authors suggest that there are only about 7 HRDDs in each country. In the United States, there are hundreds of HRDDs, and it is not clear how you came to this number, unless that is based on how many responses they received in your survey.
Answer: You are right. Our survey was intended to be sent to European brigades. One Canadian brigade has received it and completed it. To precise this information dealing with the number of dogs per country, we now name the country where the survey was filled.
Type of Manuscript: Since a discussion of the available scientific literature is part of any manuscript of original research, I think this manuscript could focus on the survey study and not attempt to describe it as a "review of the literature," since the editorial and peer review criteria for a scientific review and original research are somewhat different. The authors could still provide ample discussion of the topic in their introduction and discussion sections without classifying the manuscript as a "review."
Answer: To deal with the comments of all three referees, we have decided to keep the original structure of the manuscript, and still consider this work as a review. This choice is based on the low number of responding brigades which does not allow us to present a research paper. However, to better present and highlight the results of the survey, they are now presented at the beginning of each section. The survey allows the reader to understand the lack of communication between scientific research and working teams.
Lines 150, 245, Figure 1: The authors use the word "research" in two areas where based on context it appears, they are using this to mean "search" (the dog searching for target odor).
Answer: This mistake has been corrected in the entire manuscript
Line 26: "...police associates..." While technically this makes sense, "assets' may be a more appropriate term than "associates."
Answer: This mistake has been corrected in the entire manuscript
Lines 29, 54: The term "thank a..." is used where "due to" may be more understandable.
Answer: This mistake has been corrected in the entire manuscript
Citations: The authors tend to excessively cite review articles books, or other sources other than original research as scientific evidence to support their statements. Review articles and books should not be used in this manner, unless the authors are specifically providing commentary on trends within the topic, or commenting on the review authors' conclusions. In addition, you appear to have cited information provided as citations in the introduction or discussion section of original research papers (vs. citing the results or specific conclusions). I recommend that if you have supporting evidence that came from a previous review or introduction/discussion section citation, that you find the source of original research and cite that instead. Granted, there is little original research currently published on the topic, however as mentioned above, a true review of the scientific literature would not include review articles or books as sources of scientific evidence, although you night provide commentary on their quality or use them to direct readers to a source of compiled information on the topic. Thus, a true review of the scientific literature would highlight how much or little original research evidence is available. In this version of their manuscript, the authors are largely compiling non peer-reviewed informational publications and providing them as scientific evidence.
Answer: We apology for this. We now cite the original scientific work instead of the review, except when we aim at citing the original idea of the authors of a review.
One more appropriate way of discussing these informational publications is to caveat statements and attribute the statement as an opinion of the original author. For example (Line 168): "The working environment should define the preferred dog size and coat length. [1,19,45]" could be written as "In their book Cadaver Dog Handbook, Rebman et al advise that the working environment should define dog size and coat length because... although we found no studies on this in the available scientific literature." Then provide commentary on why they agree or disagree with the opinion of the authors of the book.
Answer: We have taken this comment into account in this new version of the manuscript.
Study Design: The survey seems to inhibit possible responses by limiting both physical and behavioral traits as well as training methods to a short list. I did not see an option for "other" as a potential response with ability to write-in responses other than "Can you describe your trainings?" which is a very broad and open question. I see this as a potential flaw in the study design. Can the authors elaborate on if the opportunity to respond outside of the provided options was available?
Answer: We make it clear now, in the material and methods, that all handlers had the possibility to add comments for each question of the survey.
You describe expanding your search criteria to include wildlife dogs starting in line 142, when a decision on study design such as this should have been described in methodology vs. Results or Discussion. You could simply state in Methods that because there would likely be more literature on these other types of dogs, and since the characteristics would be similar, you included those types of dogs in your search.
Answer: as mentioned in line 96, we extended the research to other detection dogs for specific parts of the review. The literature on wildlife detection dogs was only considered for the morphological traits (subsection “selection”) as they have to face similar environments. For other criteria (such as behavioral or olfactive traits) we took into account all literature dedicated to detection dogs. To make things clearer, we’ve added an explanation in the material and methods section (line 97-105).
Analysis of results and conclusions: I feel that the small sample size makes accurate and representative analysis difficult. Because it is unclear how many respondents you had and number of potential respondents in the target population, it is difficult to assess the overall quality of your results. Because you did get quantifiable results (number of respondents, importance score, etc.) I think should attempt a more robust statistical analysis than simply descriptive statistics.
Answer: We better describe the number of questioned handlers in the material and method section and respondents (Table 2). However, as handlers could skip a question, the effective is always different. In addition, the number of handlers who have answered each question is not sufficient to present a robust statistical analysis. In this new version, when possible, we provide chi-square tests (line155-157)
Reviewer 2 Report
The current paper outlines a study comparing a survey and a literature research about cadaver detecting dogs. The sample size is limited for both studies (N=16 teams with not really specified numbers of respondents for the survey but much fewer in case of several questions; and N=2+unknown for the literature review. However, the extraordinary nature of such subjects justifies the small sample size and the literature review is extensive and informative. Given the very limited number of such projects around the world, I consider the manuscript valuable.
Having said this, I have a lot of comments that I feel need to be addressed to get the most out of this work. Three major issues need to be addressed: 1. the structure of the manuscript, 2. the statistics of the study and 3. the writing of the manuscript (requiring copy editing to assist the expression of the authors’ ideas, e.g. present, future, and past tenses are inconsistently used, the text is full of typos). These small details add up and distract from an otherwise interesting study.
Abstract
l 14-15 Add how many responses from many countries were gathered.
Materials and methods
Add ‘Subjects’ section and describe how many papers have been collected for the literature review and how many surveys from the actual teams.
Explain why only descriptive statistics have been used. Why the authors have not compared e.g. countries, teams, etc.?
l97 Add how many papers were gathered with different bibliographic researches in Table 1.
l101 (table 2) is in fact table 1. Reference to the actual Table 2 is missing
l103 Add how many teams were contacted and calculate the response rate. Add how many teams and persons responded from how many countries. Add the language of the original survey.
l104 Table 2. Were Q2, Q4, Q8 open questions? Add how many responses were gathered per question (or indicate missing data).
l106-114 replacing the categorical responses with arbitrary numbers (0, 50, 100) are misleading and unnecessary. I recommend deleting the whole section, including the equation. Consequently, delete the scores in Fig. 2. The bars convey the information more precisely.
Results and discussion
The results of the bibliographic research and the survey are mixed and therefore this section is very difficult to follow. I recommend dividing them into two studies: 1. literature research, 2. survey, add a short discussion for each, and a general discussion at the end.
l175 The two figures in Figure 1 should be ungrouped. The sum proportion of breeds seems to below 100. Mesocephalic is more frequently used than mesaticephalic.
l179 No google search hit for "Tibor Csörg". Is this the correct name?
l275 “The response distribution describes the percentage of handlers (n=10)”. Instead of percentage the number of handlers. Explain (elsewhere) why so many responses are lacking.
l289 “It is however possible” Explain why the handlers may plan to train their dog to better resist to stressful situations if none of them mentioned this in their replies.
l300 Table 2. There is no reference in the text for Table 2. The function of this table is unclear. According to the legend it is about the “Selection methods of detection and service dogs”. Unclear what the link is between detection and service dogs as service dogs do not detect scents (rather they help e.g. disabled people). Unclear how the methods have been collected and evaluated. Who and how many persons conducted the evaluation?
What does IFT mean in the legend?
l339 types of “operant conditioning” did the authors mean “training”?
l362 Figure 4. Indicate the name of the countries. I.e. what does the pie chart on the Benelux countries represent? Is see only 10 pie charts on the maps – does this mean that the 16 teams belonged to 10 countries?
Minor comments
l15 is compared
l17 found
l21 “we enhance a wide diversity of…” – unclear
l29 and l 54 thanks to
l176 German shepherd dog
l182 whatever the odour source is
l223 delete ‘are’
l238 from how many countries?
l238 assessed
l265 did evaluate
l268 What ‘They’ refer to?
l302 (Table 2) ability to stay relaxed
mark of attention to the owner – sign instead of mark?
l305 delete These
l306 service dogs
l307 were instead of are
l409 olfactory
Author Response
The current paper outlines a study comparing a survey and literature research about cadaver detecting dogs. The sample size is limited for both studies (N=16 teams with not really specified numbers of respondents for the survey but much fewer in case of several questions; and N=2+unknown for the literature review. However, the extraordinary nature of such subjects justifies the small sample size and the literature review is extensive and informative. Given the very limited number of such projects around the world, I consider the manuscript valuable.
Answer: we provided more information in the material and methods in regard to the number of handlers answering the survey.
Having said this, I have a lot of comments that I feel need to be addressed to get the most out of this work. Three major issues need to be addressed: 1. the structure of the manuscript, 2. the statistics of the study and 3. the writing of the manuscript (requiring copy editing to assist the expression of the authors’ ideas, e.g. present, future, and past tenses are inconsistently used, the text is full of typos). These small details add up and distract from an otherwise interesting study.
Answer: We revised the structure of the manuscript as presented in the new submission. Statistical analyses were also performed when it was possible. Finally, the tenses through the document are now homogenous. We use past tense to describe the materials and methods performed for the bibliographic research and for the survey. Regarding general rules, habits and results validated in the literature, we use the present. We use the future to describe perspectives.
l 14-15 Add how many responses from many countries were gathered.
Answer: We’ve added this information
Explain why only descriptive statistics have been used. Why the authors have not compared e.g. countries, teams, etc.?
Answer: we’ve added quantitative statistics when possible (line 165). We are not able to provide a quantitative statistic to compare countries or brigades for two main reasons. First, the number of handlers answering the survey is too small. Second, depending on the brigades, sometimes each handler of a brigade answered the survey, sometimes only the chief of the brigades answered the survey. The statistics would be too variable (Normality, homoscedasticity …)
l101 (table 2) is in fact table 1. Reference to the actual Table 2 is missing
Answer: Correction is made
l103 Add how many teams were contacted and calculate the response rate. Add how many teams and persons responded from how many countries. Add the language of the original survey.
Answer: All contacted cadaver searching brigades have answered the survey. This is the result of the fact that the chief of KINOPOL has sent the survey to all cadaver brigades. We have added information in the M&M section (country, number of brigades, the language of the survey...)
l104 Table 2. Were Q2, Q4, Q8 open questions? Add how many responses were gathered per question (or indicate missing data).
Answer: Yes, these were open questions. we provide the number of answers received in de legend.
l106-114 replacing the categorical responses with arbitrary numbers (0, 50, 100) are misleading and unnecessary. I recommend deleting the whole section, including the equation. Consequently, delete the scores in Fig. 2. The bars convey the information more precisely.
Answer: To meet all reviewers’ recommendations we have decided to keep this presentation of the results.
The results of the bibliographic research and the survey are mixed and therefore this section is very difficult to follow. I recommend dividing them into two studies: 1. literature research, 2. survey, add a short discussion for each, and a general discussion at the end.
Answer: To meet all reviewer recommendations, we highlight the results of the survey at the beginning of each subsection.
l175 The two figures in Figure 1 should be ungrouped. The sum proportion of breeds seems to below 100. Mesocephalic is more frequently used than mesaticephalic.
Answer: Correction is made
Reviewer 3 Report
The subject of the manuscript is very interesting and focuses on specific issues. The work was divided into chapters and subchapters forming a logical whole. It is both a review and research work. This is a very extensive work, referring to many current literature items.
My comments relate to the following issues:
In my opinion, modification of the title should be considered. I suggest "Selection and Training Methods of Cadaver Dogs - a survay among handlers and literature review" or another, in a similar style. "Deathly Hallows" do not seem to match.
Line 57: In my opinion, the authors should explain how the size of the dog affects his predisposition to olfactory work. In the further part of the work it is indeed mentioned that the smaller size of the dog is the smaller the area of the olfactory epithelium. At this point in the manuscript, however, the reader does not know properly how the size of the dog affects his predisposition to olfactory work. Are small sized dogs or big sized dogs excluded from olfactory work?
Table 1, line 98: In my opinion, it is not necessary to describe the methodology of literature review so precisely. Table 1 is not needed.
Line 104: Table 1 again? The numbering of the tables must be corrected.
Line 171: Reference to figure 1 in my opinion is not at this point as it should. The reference to figure 1 should be in the place where the authors describe the breeds that should be preferred by handlers.
Authors should think about how to separate literature review results from survey results. The results of surveys conducted among handlers are very significant, and they get lost in the literature review. I think that it would be worth to emphasize them more.
line 293: iv ? wrong section numbering?
line 294: Reference to table 3, which is called table 2. It is necessary to organize the numbering of the tables
line 313-320: Sounds like conclusion to me.
Conclusions
I suggest you change the form of the conlusions chapter. The authors write what they suggest to validate / develope etc. In my opinion it would be worth to point out that the surveys and literature review results show that this is lacking of this. And then suggest what to do.
for example: "There is lack of selection methods based on anatomical, olfactive and behavioral traits" instead of "A selection method based on anatomical, olfactive and behavioral traits should be developed"
Author Response
Referees comments in normal type – Our comments in bold
We thank the referee for the careful reading and helpful comments
The subject of the manuscript is very interesting and focuses on specific issues. The work was divided into chapters and subchapters forming a logical whole. It is both a review and research work. This is a very extensive work, referring to many current literature items. In my opinion, modification of the title should be considered. I suggest "Selection and Training Methods of Cadaver Dogs - a survey among handlers and literature review" or another, in a similar style. "Deathly Hallows" do not seem to match.
Answer: We have selected this title as a reference to the books and movies “Harry Potter and the deathly hallows”. We would love to keep this title, as it is both informative as well as catching someone’s eye.
Line 57: In my opinion, the authors should explain how the size of the dog affects his predisposition to olfactory work. In the further part of the work it is indeed mentioned that the smaller size of the dog is the smaller the area of the olfactory epithelium. At this point in the manuscript, however, the reader does not know properly how the size of the dog affects his predisposition to olfactory work. Are small sized dogs or big sized dogs excluded from olfactory work?
Answer: To make things clearer, we decided to only focus on parameters that impact the olfaction of detection dogs in this paragraph. So we replace dog size by head conformation (line 57).
Table 1, line 98: In my opinion, it is not necessary to describe the methodology of literature review so precisely. Table 1 is not needed.
Answer: Since publications dedicated to HRDDs are not numerous, we had to extend the literature review to other detection dogs. In our opinion, it is therefore better to keep table 1 to make things clearer. It is also a recommendation of the other two referees to clarify our methodology.
Line 104: Table 1 again? The numbering of the tables must be corrected.
Answer: Corrected: changed to table 2
Line 171: Reference to figure 1 in my opinion is not at this point as it should. The reference to figure 1 should be in the place where the authors describe the breeds that should be preferred by handlers. Authors should think about how to separate literature review results from survey results. The results of surveys conducted among handlers are very significant, and they get lost in the literature review. I think that it would be worth to emphasize them more.
Answer: we emphasized the results of the survey at the beginning of each section. The place of each figure therefore had to be changed.
line 293: iv ? wrong section numbering?
Answer: Corrected
line 294: Reference to table 3, which is called table 2. It is necessary to organize the numbering of the tables
Answer: Corrected
line 313-320: Sounds like conclusion to me.
Answer: Indeed, we decided to make a short conclusion at the end of each section (selection and training) and to summarize them at the end of the paper.
I suggest you change the form of the conlusions chapter. The authors write what they suggest to validate / develope etc. In my opinion it would be worth to point out that the surveys and literature review results show that this is lacking of this. And then suggest what to do.for example: "There is lack of selection methods based on anatomical, olfactive and behavioral traits" instead of "A selection method based on anatomical, olfactive and behavioral traits should be developed"
Answer: the conclusion was reformatted following your advice.
Round 2
Reviewer 1 Report
Overall this is a much-improved version of the manuscript which is easier to follow and is more focused on one goal (review format). I feel now that there are only a minor edits needed regarding English grammar and punctuation. I appreciate the author's extensive effort to reformat and re-write their manuscript in response to our comments.
Author Response
Small changes have been provided after reading the manuscript again.
Reviewer 2 Report
Thank you for the corrections. Before approving the manuscript I suggest the following minor changes:
l121 "the words canine team are dedicated to the couple handler-HRDD." In fact, the words 'canine team' are used only once more in the text (l189). 'Handlers' are used throughout the text, therefore defining 'canine team' seems to be unnecessary.
l189 canine teams (over 50). Above it was claimed that N(handlers)=50 and not more than 50.
l162 "All Anglo-Saxon dogs are neutered" Please list the countries instead of this expression.
l212 " unfortunately, size and coat length were not being considered to..." Why is this fact "unfortunate"?
l259-260 "Personality is functionally indistinguishable from temperament" The distinction between temperament and personality is not used consistently in the literature (see Jones and Gosling, 2005, Definitions of temperament and personality section). Therefore I suggest deleting the "is functionally indistinguishable from temperament" part of the sentence and provide the personality definition only.
l339 year (or reference number) is missing after Brownell and Marsolais
Author Response
Thank you for the corrections. Before approving the manuscript I suggest the following minor changes:
l121 "the words canine team are dedicated to the couple handler-HRDD." In fact, the words 'canine team' are used only once more in the text (l189). 'Handlers' are used throughout the text, therefore defining 'canine team' seems to be unnecessary.
=> Line 121 has been deleted. We replaced 'canine team" with 'handler', as recommended.
l189 canine teams (over 50). Above it was claimed that N(handlers)=50 and not more than 50.
=> 'over 50' has been replaced by 'out of 50'.
l162 "All Anglo-Saxon dogs are neutered" Please list the countries instead of this expression.
=> 'All anglo-saxon dogs" has been replaced by 'All Canadian and English dogs'.
l212 " unfortunately, size and coat length were not being considered to..." Why is this fact "unfortunate"?
=> This sentence has been changed to: 'While agility was classified as the most important trait, size and coat length were not being considered by the international teams having answered our survey when selecting HRDDs".
l259-260 "Personality is functionally indistinguishable from temperament" The distinction between temperament and personality is not used consistently in the literature (see Jones and Gosling, 2005, Definitions of temperament and personality section). Therefore I suggest deleting the "is functionally indistinguishable from temperament" part of the sentence and provide the personality definition only.
=>We have adapted the sentence following this advice
l339 year (or reference number) is missing after Brownell and Marsolais
=> The year has been aded